**www.cambridge.org/qrd**

## Perspective

Molecular Dynamics simulations; Molecular Docking; RNA selectivity; RNA-binding drugs; non-coding RNA

**Author for correspondence:**
*Mattia Bernetti,
E-mail: mattia.bernetti@iit.it;
Andrea Cavalli,
E-mail: andrea.cavalli@unibo.it

# Computational drug discovery under RNA times

Mattia Bernetti[1,2]* , Riccardo Aguti[1,2] , Stefano Bosio[1,2] ,
Maurizio Recanatini[2] , Matteo Masetti[2] and Andrea Cavalli[1,2]*

[1]Computational and Chemical Biology, Italian Institute of Technology, 16152 Genova, Italy and [2]Department of Pharmacy and Biotechnology, Alma Mater Studiorum-University of Bologna, 40126 Bologna, Italy

### Abstract

RNA molecules play many functional and regulatory roles in cells, and hence, have gained considerable traction in recent times as therapeutic interventions. Within drug discovery, structure-based approaches have successfully identified potent and selective small-molecule modulators of pharmaceutically relevant protein targets. Here, we embrace the perspective of computational chemists who use these traditional approaches, and we discuss the challenges of extending these methods to target RNA molecules. In particular, we focus on recognition between RNA and small-molecule binders, on selectivity, and on the expected properties of RNA ligands.

### Introduction

RNA's biological relevance has traditionally been ascribed to its role as an intermediate in the flow of genetic information from DNA to the production of functional proteins. However, it has become increasingly evident that RNA has many functional and regulatory roles in all domains of life. For instance, RNA molecules can regulate gene expression directly or by interacting with small organic molecules (miRNA, Pasquinelli *et al.,* 2005; riboswitches, Serganov and Nudler, 2013, respectively), and can exert enzymatic activity (ribozymes, Doudna and Cech, 2002). Therefore, being involved in many cellular pathways, RNA molecules offer opportunities as targets for the development of therapeutic strategies (Shortridge and Varani, 2015; Connelly *et al.,* 2016; Matsui and Corey, 2017; Warner *et al.,* 2018; Falese *et al.,* 2021). However, most of the drug discovery efforts have focused on proteins, which have long been known to modulate cellular activity. Nevertheless, in the human genome only a small fraction of the transcribed RNA is translated into proteins (Fig. 1) (Warner *et al.,* 2018; ENCODE Project Consortium, 2012; Oliver *et al.,* 2020), and only a small portion of these proteins has been successfully targeted (Warner *et al.,* 2018). The growing characterisation of noncoding RNA, therefore, offers new opportunities for novel therapeutic approaches, which may be particularly valuable when seeking alternatives in cases of drug resistance or when traditionally undruggable protein targets are encountered.

Computational approaches have reached the status of standard tools in drug discovery campaigns (Macalino *et al.,* 2015). This is particularly true for structure-based drug design (Jorgensen, 2004), where the macromolecule's structural information is used to find small molecules able to bind and modulate its activity. These computer-aided approaches have mostly been used to identify drugs for protein targets, as RNA has only recently been fully recognised as a relevant pharmaceutical target. Moreover, RNA targets are particularly challenging for standard computational approaches due to their complex structural dynamics and high charge density. Indeed, the applicability of standard approaches to RNA targets is still being debated and is a hot topic for research (Fedorova *et al.,* 2018; Warner *et al.,* 2018; Juru and Hargrove, 2021; Manigrasso *et al.,* 2021).

In this Perspective article, we put ourselves in the shoes of computational medicinal chemists who have experience in targeting proteins and who wish to extend their expertise to RNA to find potent and selective small-molecule ligands. We discuss critical aspects associated with this transition, which may require additional operations or some rethinking of standard procedures. The Perspective is divided into three sections that reflect the main challenges of computational RNA-targeted drug discovery (Fig. 2). First, we address RNA-small molecule recognition, particularly how RNA dynamics should be treated and how to identify potential binders. Second, we address small-molecule selectivity for RNA targets. Third, we address RNA binders and their expected physicochemical properties. We conclude with an outlook on future opportunities for computational medicinal chemists on the way to a wider playground, where to apply and expand their expertise.

### RNA-small molecule recognition

Rational drug discovery campaigns typically start with a biomolecular target that has been pre-clinically validated. Validated targets play a critical role in a physio-pathological process and their

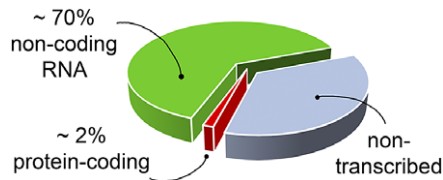

**Fig. 1.** The targetable portions of the human genome. More than 70% of the human genome is transcribed into RNA, but only a small portion of this encodes for, and is thus translated into, proteins (red slice) (ENCODE Project Consortium, 2012; Oliver *et al.*, 2020), of which, only a small fraction has been successfully targeted with drugs (Warner *et al.*, 2018). The possibility of targeting non-coding functional RNA molecules (green slice) could significantly increase the number of drug discovery strategies.

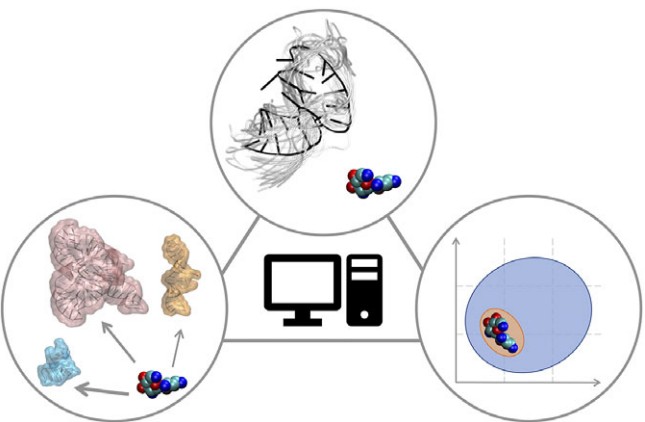

**Fig. 2.** RNA-targeted computational drug discovery. A schematic representation of the Perspective's three main sections: structural dynamics of the target in RNA-ligand recognition (top), target selectivity (left) and physicochemical properties of RNA binders (right).

modulation is likely to produce a therapeutic effect, hopefully within an acceptable safety window. Drug discovery pipelines, therefore, begin by identifying small-molecule hits for a validated target (Hughes *et al.*, 2011). In a structure-based context, computational strategies for this stage include fragment-based approaches (Erlanson *et al.*, 2016), de novo design (Schneider and Fechner, 2005), and, most importantly, virtual screening of small-molecule libraries (Maia *et al.*, 2020). In a virtual screening campaign, a small-molecule library undergoes molecular docking, a procedure that aims at predicting how the ligands bind to the target. This computational approach is rapid and can be used to roughly discriminate between binders and non-binders. As such, it has become a well-established strategy for the identification of small-molecule hits.

There are two particularly critical aspects involved in extending docking protocols, that have long been refined over proteins, to RNA targets: i) how to describe the target's structure, and in particular how to account for its intrinsic flexibility and structural dynamics, and ii) how to assess quantitatively the binding poses. These aspects are already critical in the context of protein targets but may be even more crucial for RNA targets. Concerning the former aspect, any docking campaign requires the target's structure as the starting point. Experimental methods to reconstruct the 3D structures at the atomistic level include X-ray crystallography, nuclear magnetic resonance (NMR), and cryo-electron microscopy (Palamini *et al.*, 2016). In the absence of experimental data, modelling approaches can be taken advantage of to reconstruct the target's structure with varying degrees of accuracy, depending on

the available information (Hameduh *et al.*, 2020). In this respect, the machine-learning based approach AlphaFold (Jumper *et al.*, 2021) is achieving impressive results in protein 3D-structure prediction, however, an equivalent for RNA does not exist yet. For protein targets, X-ray crystallography experiments have long been established as an efficient method to produce high-resolution structures. Therefore, a docking effort on a protein target typically begins with a crystal structure. In contrast, a significant fraction (about 40%) of RNA structures is solved via NMR (Barnwal *et al.*, 2017), while crystal structures can be often found in the cases of larger and structurally complex RNA molecules.

While a reliable structure is certainly a great starting point, it may however not suffice for a comprehensive description of the target of interest. Indeed, biomolecules are not frozen entities, but rather present in solution various degrees of structural flexibility. This dynamic nature not only can influence the binding of molecular partners, but structural modifications can also be triggered upon the binding of smaller molecules. Therefore, including information about the dynamics of the target during a docking procedure can depict more realistically what really occurs at the molecular level and potentially improve results. This is a long-standing issue in computational drug discovery (Feixas *et al.*, 2014). Indeed, protein flexibility is typically addressed in docking protocols through different approaches (Buonfiglio *et al.*, 2015), such as soft-docking (Ferrari *et al.*, 2004) and induced-fit docking (Sherman *et al.*, 2006). These strategies can be considered an integral part of the docking protocol. While they do not usually affect the performance of the docking calculation much, they do account for a rather limited structural flexibility of the target. An alternative approach is to include the receptor dynamics as an ensemble of multiple conformations (Huang and Zou, 2007; Amaro *et al.*, 2018). This strategy allows greater conformational changes of the target, but the docking calculation scales up linearly as the procedure must be iterated for each of the structures in the ensemble (Fig. 3).

From a practical standpoint, the ensemble is generated separately from the docking calculation using other computational methods. This ensemble docking approach has been used for protein targets (Amaro *et al.*, 2018). Compared to common protein targets, RNA molecules display marked and complex structural dynamics (Ganser *et al.*, 2019). Therefore, the ensemble docking approach appears as the natural choice and should indeed be preferred for RNA targets. The literature contains some successful examples in this direction (Stelzer *et al.*, 2011; Ganser *et al.*, 2018).

A diverse set of computational approaches can be used to generate RNA conformational ensembles. In particular, static frameworks could be employed to generate a pool of RNA structures, as it is done through the popular Fragment Assembly of RNA with Full-Atom Refinement (FARFAR) algorithm in the Rosetta software suite (Watkins *et al.*, 2020). In contrast, methods that mimic the dynamics, such as molecular dynamics (MD) simulations, can be employed in the generation of a conformational ensemble (Sponer *et al.*, 2018). MD explores the conformational dynamics of biomolecules under realistic conditions (e.g., explicit solvent, quasi-physiological ionic concentrations) and has become an indispensable tool for investigating mechanistic features at the atomistic level (De Vivo *et al.*, 2016; Decherchi and Cavalli, 2020). Notably, the results of MD simulations strongly depend on the ability of the underlying model (i.e., the force field) to capture the physics of the interactions in molecular systems. Since structural biology and drug discovery have long been focused on proteins, force fields for RNA have developed at a much slower pace (Table 1).

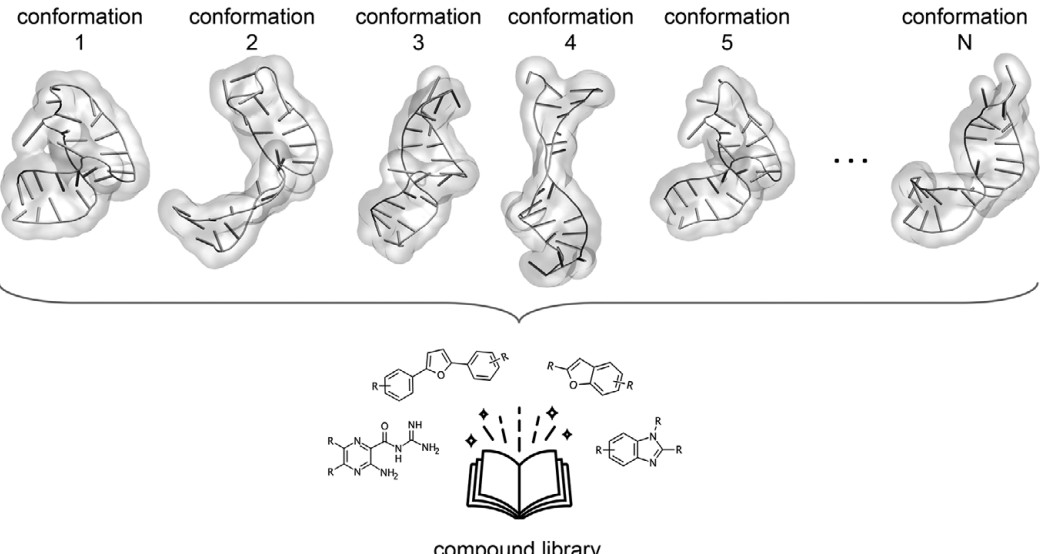

**Fig. 3.** Ensemble docking. An ensemble comprising multiple conformations of the target is included to take into account its structural dynamics. The docking calculation (virtual screening for large libraries) is repeated for each structure in the ensemble. The RNA structures here belong to the conformational ensemble of the transactivation response element (TAR) RNA from human immunodeficiency virus type-1 reconstructed in Salmon *et al.* (2013).

Among the Amber family of force fields, the ff99 force field combined with the bsc0 and $\chi_{OL3}$ refinements is considered state-of-the-art, as it is the most validated and widely used (Wang *et al.,* 2000; Pérez *et al.,* 2007; Zgarbová *et al.,* 2011). In particular, ff99 (Wang *et al.,* 2000) is a major branch of Amber force fields that includes parameters for both proteins and nucleic acids and was based on the previous ff94 version (Cornell *et al.,* 1995). For DNA and RNA, the main difference between ff94 and ff99 is the refinement of the sugar puckering and χ dihedral parameters. In 2007, Orozco and coworkers introduced a major correction, known as bsc0, where the α and γ dihedral angles of the nucleic acid backbone were modified to avoid the formation of nonnative γ-trans backbone dihedral states, thus reducing unrealistic helical twists in A-RNA (Pérez *et al.,* 2007). In 2011, the $\chi_{OL3}$ refinement involved a reparameterisation of the χ dihedral to prevent high-*anti* γ shifts in RNA, which led to entirely untwisted and ladder-like structures (Zgarbová *et al.,* 2011). In more recent attempts to improve the force field, dihedral reparameterisation was conducted in the Mathews group (Aytenfisu *et al.,* 2017), while dihedral, electrostatic and van der Waals parameters were considered by the Shaw group (Tan *et al.,* 2018). Recently, different schemes were also proposed, which used an additional term to better describe hydrogen-bond interactions (Fröhlking *et al.,* 2022) or introduced the grid-based energy correction map (CMAP) term in the context of RNA force fields (Chen *et al.,* 2022). The ability of the CHARMM and OPLS families of force fields to describe RNA structures is also gradually being improved (see Table 1) (Denning *et al.,* 2011; Robertson *et al.,* 2019). In addition to the force field, the thoroughness of conformational sampling is a common issue for both protein and RNA modelling. This is because the timescales that are accessible via conventional (or "plain") MD simulations are still limited (in the order of tens of microseconds for most of the research projects). Nevertheless, outstanding results have been achieved thanks to hardware advances (Pande *et al.,* 2003; Shaw *et al.,* 2014) and sophisticated enhanced sampling approaches (Abrams and Bussi, 2013; Mlýnský and Bussi, 2018) being implemented in popular instruments such as PLUMED (Tribello *et al.,* 2014). In the context of proteins, particularly challenging systems in terms of conformational sampling are

intrinsically disordered proteins (Habchi *et al.,* 2014). In this respect, enhanced sampling methods were successfully employed (Granata *et al.,* 2015; Palazzesi *et al.,* 2015; Bernetti *et al.,* 2017; Masetti *et al.,* 2020), and we thus envision that they will find increasing application also on RNA molecules.

Despite recent improvements, the current force fields still bear some limits. Combined with the shortcomings linked to the sampling, they can generate RNA conformational ensembles that do not entirely agree with experiments. While this is problematic in general, it may become particularly critical when using the generated structural ensembles for docking. However, experimental information can be used to generate more reliable ensembles (Pitera and Chodera, 2012; Hummer and Köfinger, 2015; Bonomi *et al.,* 2016, 2017; Cesari *et al.,* 2018; Orioli *et al.,* 2020) in both static (Shi *et al.,* 2020) and dynamic frameworks (Bottaro *et al.,* 2018). During MD simulations, for example, experimental data can be included on the fly to guide the sampling towards regions of the conformational space that are supported by experiments. Alternatively, reweighting approaches can be applied after the MD simulation to identify those conformations that better agreed with the experimental data. Both strategies have been successfully applied to reconstruct reliable conformational ensembles of RNA molecules (Borkar *et al.,* 2013; Bottaro *et al.,* 2018). Notably, the experimental data here can come from a broader range of sources than the data on the initial atomistic structures. Indeed, any experimental information that can be related to an observable computed from the biomolecule coordinates in the MD trajectory can be exploited. Thus, experimental methods that provide coarser structural information (e.g., small-angle X-ray scattering, SAXS) are extremely valuable and have been used for this purpose (Bernetti *et al.,* 2021). Finally, clustering algorithms that have become of routine use in the context of MD simulations (Bernetti *et al.,* 2020) can be of remarkable support to select representative structures from MD-generated ensembles for the subsequent docking/virtual screening stage.

Given a biomolecular target's structure or ensemble of structures, the docking procedure attempts to find plausible ligand-target bound configurations, that is, binding modes (or "poses")

**Table 1.** Main classes of RNA force fields and their major variants

| Year | FF name | Composition | Main features |
|------|---------|-------------|---------------|
| | | Amber | |
| 1995 | ff94 (Cornell *et al.,* 1995) | ff94 | |
| 1999 | ff98 (Cheatham *et al.,* 1999) | ff94 + P + $\chi$ | Improves pucker and twist; comparable to ff94 |
| 2000 | ff99 (Wang *et al.,* 2000) | ff98 + P | Improves pucker; small changes |
| 2007 | parmbsc0 (Pérez *et al.,* 2007) | ff99 + $\alpha\gamma_{bsc0}$ | Avoids nonnative $\alpha/\gamma$ conformations but penalises native $\gamma$ ones |
| 2011 | ff99bsc0$\chi_{OL3}$ (Zgarbová *et al.,* 2011) | ff99 + $\alpha\gamma_{bsc0}$ + $\chi_{OL3}$ | Prevents high-*anti* g shifts; state-of-the-art force field for amber |
| 2010 | Amber99$\chi$ (Yildirim *et al.,* 2010) | ff99 + $\chi_{YIL}$ | Reduces ladder-like structures and the A-form inclination |
| 2012 | Amber99TOR (Yildirim *et al.,* 2012) | ff99 + $\chi_{YIL}$ + $\beta,\epsilon,\zeta_{YIL}$ + $\alpha\gamma_{bsc0}$ | Improves the description of cytidine and uridine in solution; performs suboptimally for canonical RNA |
| 2017 | Aytenfisu−Spasic−Stern−Mathews (Aytenfisu *et al.* (2017) | ff99 + $\alpha\beta\gamma\epsilon\zeta\chi_{Mathews}$ | Performs well for tetraloops; reduces intercalation events |
| 2013 | Chen−Garcia (Chen and García (2013) | ff99 + vdW$_{GC}$ + $\chi_{GC}$ | Reduces stacking, but overstabilises hydrogen-bond interactions between bases |
| 2018 | Tan−Piana−Dirks−Shaw (Tan *et al.* (2018) | ff99bsc0$\chi_{OL3}$ + $\gamma\zeta\chi_{Shaw}$ + vdW$_{Shaw}$ + electrostatics$_{Shaw}$ | Focused on non-bonded interactions; the conformational ensembles closely reproduce experimental ones |
| 2022 | gHBfix21 (Fröhlking *et al.,* 2022) | ff99bsc0$\chi_{OL3}$ + HbFix | Improves hydrogen-bond interactions, stabilises native structures |
| 2022 | ff99OL3_CMAP1 (Chen *et al.,* 2022) | ff99bsc0$\chi_{OL3}$ + $\zeta/\alpha_{CMAP}$ | Decreases population of incorrect structures; improves stability of tetranucleotides |
| | | CHARMM | |
| 1995 | CHARMM22 (MacKerell *et al.,* 1995) | CHARMM22 | |
| 2000 | CHARMM27 (MacKerell *et al.,* 2000) | CHARMM27 | Significant improvements over CHARMM22; known issues with pair opening |
| 2011 | CHARMM36 (Denning *et al.,* 2011) | CHARMM27 + 2'-OH dihedral | Partial stabilisation of the structures; state-of-the-art force field for CHARMM |
| | | OPLS | |
| 1991 | OPLS-AA (Pranata *et al.,* 1991) | OPLS-AA | |
| 2019 | OPLS-AA/M (Robertson *et al.,* 2019) | OPLS-AA + $\alpha\gamma\chi$ + P | Reduces intercalation events and is well-suited to describing non-canonical motifs |

with overall favourable interactions between the two binding partners. We here briefly outline some popular docking software packages and refer the interested reader to a recent comprehensive overview (Zhou *et al.,* 2021). Glide (Friesner *et al.,* 2004), GOLD (Jones *et al.,* 1997) and AutoDock Vina (Trott and Olson, 2010) are software packages that were devised for protein targets and that can also be used for RNA when needed. Conversely, AutoDock (Morris *et al.,* 1998), DOCK 6 (Lang *et al.,* 2009) and ICM (Abagyan *et al.,* 1994) have been improved to be used with RNA through dedicated docking protocols, optimised ligand-sampling algorithms, or the inclusion of solvation effects either in the generation of poses or in the scoring functions. Finally, the software packages MORDOR (Guilbert and James, 2008), rDOCK (Ruiz-Carmona *et al.,* 2014), and the very recent RLDOCK (Sun *et al.,* 2020) and NLDock (Feng *et al.,* 2021) were specifically developed for RNA docking, reflecting the growing interest in RNA-oriented drug discovery. The trend that emerges from the validation of the latter approaches on diverse datasets of experimental RNA-ligand complexes, usually evaluated as the success rate in reproducing experimental binding poses, is

that RNA-specific methods outperform the tools developed for proteins or generic macromolecules (Feng *et al.,* 2021; Zhou *et al.,* 2021). Notably, such trend highlights the relevance of specifically considering interaction and structural features that are peculiar of RNA-ligand binding.

The quality of the poses identified by the docking procedure can be assessed through a variety of strategies that fall under the term "scoring functions". These can be broadly classified as the follows: i) knowledge-based, when the scoring is built upon information extracted from known three-dimensional structures of target-ligand complexes; ii) physics-based, when the scoring is based on force fields or simplified empirical functions of the target-ligand interactions and iii) machine learning (ML)-based, when the scoring is evaluated through ML models trained on available experimental data (Zhou *et al.,* 2021). Scoring functions are typically included in docking software. However, there has been a rapid growth in standalone options, including the knowledge-based ITScore-NL scoring function (Feng and Huang, 2020) and the ML-based RNAPosers (Chhabra *et al.,* 2020), RNAmigos (Oliver

et al., 2020) and AnnapuRNA (Stefaniak and Bujnicki, 2021) scoring functions. Notably, MD simulations can also be used for scoring (Menchon *et al.,* 2018). Indeed, binding pose stability can be assessed with plain MD runs or, alternatively, binding affinities can be estimated with more computationally intensive MD-based free energy calculations (Decherchi and Cavalli, 2020). These procedures can straightforwardly be applied to RNA targets. The only caveats are the accuracy of the force field and, most importantly, the limited number of RNA-ligand complexes that can typically be managed via these approaches, which hinders their use for large-scale virtual screenings.

## Selectively targeting RNA structures

Most functional proteins targeted in drug discovery campaigns fold in a well-defined native state under physiological conditions. The functional activity of proteins usually takes place in surface cavities. In structure-based approaches, structural information about these cavities is directly used to design ligands that might bind there (Pérot *et al.,* 2010). If the target of interest is well-characterised, relevant binding sites may already be known. Often, however, either the binding site is unknown or alternative binding sites may be sought (e.g., for allosteric modulation) (Kuzmanic *et al.,* 2020). The possibility of identifying binding pockets by computational means is thus integral to modern structure-based drug design. The Site-Map tool of the Schrödinger suite (Halgren, 2009), the ICM Pocket-etFinder (An *et al.,* 2005), the NanoShaper software suite (Decherchi and Rocchia, 2013) and its dynamic extension Pocketron (La Sala *et al.,* 2017) are popular options in this regard (Pérot *et al.,* 2010). These tools have been widely employed to detect pockets in proteins. Although some of them have also been used to identify pockets in RNA molecules (Ganser *et al.,* 2018; Hewitt *et al.,* 2019; Panei *et al.,* 2022), an extensive exploration of their applicability is still missing. In proteins, suitable binding pockets have a well-defined 3D organisation of the amino acids comprised therein, which, alongside their composition and variability, display a wealth of physicochemical features (e.g., balance between hydrophobic/hydrophilic regions, solvent exposure and shape) that make the pockets rather distinctive. Taken together, these aspects encode, to a certain extent, the target selectivity that can potentially be achieved by addressing that site (Ehrt *et al.,* 2016; Smilova *et al.,* 2022). Usually, however, this information is not extensively exploited in the early stages of the drug discovery pipeline, where most of the effort is directed towards the identification of potential hits. Conversely, it is only during later phases of lead optimisation that target selectivity is fully explored, together with chemical modifications that can improve the affinity and ADMET (absorption, distribution, metabolism, excretion and toxicity) properties.

Moving into the context of RNA, molecules with functional roles in cellular pathways display a markedly heterogeneous structural complexity. Indeed, functional RNA molecules vary from short hairpin loops and miRNA, to tRNA, riboswitches and larger ribozymes, up to the scale of the ribosome (Ganser *et al.,* 2019). From a structure-based drug discovery perspective, we can broadly distinguish two main scenarios depending on the complexity of the RNA target's molecular structure.

In the first scenario, relatively simple secondary structure elements can be identified as hot spots for small-molecule binding. These short stem-loop motifs include for instance apical loops, bulges and internal loops, and have been extensively studied in this regard (Liu *et al.,* 2004; Disney and Childs-Disney, 2007; Meyer and Hergenrother, 2009). Here, the structure-based strategy is made

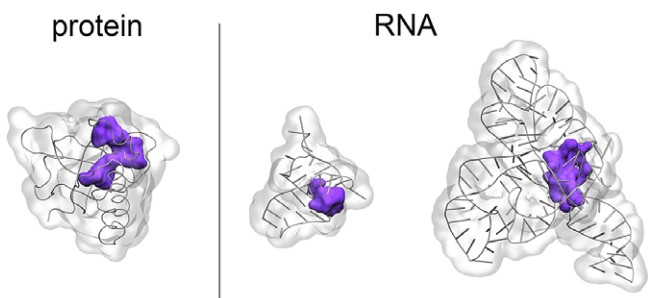

**Fig. 4.** Protein and RNA binding pockets. Binding pockets in proteins (left, riboflavin kinase, PDBID: 1NB9) are typically highly structured. In RNAs, structured pockets like those in proteins are found in highly folded structures (right, FMN riboswitch, PDBID: 3F4G). In contrast, relatively simple RNA structures (middle, HIV TAR, PDBID: 1QD3) usually offer shallow or relatively small pockets, which are more challenging to target with small molecules. The pockets shown herein (in violet) were identified with the NanoShaper software (Decherchi and Rocchia, 2013). The two RNA structures were chosen as representatives of good and intermediate quality pockets from the examples reported by Warner *et al.* (2018): in this work, pocket quality was estimated using the ICM tool PocketFinder (An *et al.,* 2005), where pockets of larger size and buriedness resulted in higher quality.

possible by the unpaired nucleobases, which can arrange in characteristic structures and thus offer the possibility of small-molecule binding (Juru and Hargrove, 2021). However, their pockets are usually shallow or relatively small, which makes it challenging to find high-affinity binders that produce specific interactions (Fig. 4) (Warner *et al.,* 2018; Juru and Hargrove, 2021). Indeed, shallow pockets pose a biophysical limit to how potent non-irreversible binders can be. Furthermore, this strategy may prove particularly arduous in terms of selectivity because similar secondary structure elements can be found across diverse RNA molecules. Indeed, only modest potency and selectivity have been achieved in reported studies focused on these types of RNA molecules (Warner *et al.,* 2018). However, this strategy may be particularly valuable and viable in the absence of a more complex tertiary structure, which offers more characteristic architectures for potential drug binding (Juru and Hargrove, 2021).

The second scenario involves RNA molecules achieving a higher level of folding and thus complex 3D structures. Although RNA molecules have less chemical variety than proteins (4 nucleotides vs 21 amino acids), this complex folding can nevertheless produce distinctive cavities reminiscent of protein-like binding pockets (Warner *et al.,* 2018; Hewitt *et al.,* 2019). These cavities can be suitable for computational approaches already established in the context of protein targets. In such cases, the determinants of RNA-ligand recognition are likely to be similar to those of protein-ligands, with no need to develop alternative RNA-centric approaches (Fedorova *et al.,* 2018). Riboswitches and ribozymes are remarkable examples of highly folded RNA molecules with complex tertiary structures (Zafferani and Hargrove, 2021). Since those interact with metabolites or substrates, they are already prone to ligand recognition in pre-formed binding pockets. Multi-junctions and pseudoknots, in general, have also been suggested as promising RNA species for RNA-targeted drug discovery because their great structural complexity is suitable for pocket formation (Warner *et al.,* 2018). Despite this, to the best of our knowledge, the literature contains just one report of a successful campaign of rational drug design using classical, established medicinal chemistry protocols (Fedorova *et al.,* 2018). This work used high-throughput experimental assays to identify hits. However, computational approaches could also be employed for hit

identification of these RNA molecules, so compound libraries could potentially be investigated on a much larger scale.

Given this overall picture, at variance with protein targets, the discourse around selectivity becomes more urgent already at an earlier stage when targeting RNA molecules. This is because the choice of a certain class of RNA targets can greatly impact the level of selectivity that can be achieved. While the second scenario discussed for RNA appears to hold more promise for identifying selective binders to modulate RNA activity, this may however preclude opportunities to develop effective drugs for pathological conditions mediated by structurally simpler RNAs (Juru and Hargrove, 2021). We, therefore, encourage researchers to be open to both scenarios while considering their respective implications for selectivity.

## Properties of RNA ligands

To identify drug candidates, it is essential to know the physicochemical features that ligands should possess in order to bind to a particular class of biomolecular targets. In computational drug discovery, this knowledge can be instrumental to design libraries for virtual screening or to guide the lead optimisation stage of a candidate. For proteins, drug discovery usually aims to identify small organic molecules with physicochemical profiles that meet the criteria of oral drugs, including solubility, bioavailability, cell and tissue permeability, chemical stability and absence of toxicity. In this respect, Lipinski's rule of five is the established guiding principle for rational drug design (Lipinski, 2004). Indeed, through a retrospective analysis of approved drugs and drug candidates, Lipinski's rule of five empirically set the drug-likeness boundaries for physicochemical parameters including molecular weight, lipophilicity and number of hydrogen-bond donors and acceptors. This, in conjunction with the wealth of knowledge accumulated through years of experience in the field, has led to the identification of a rather defined chemical space that is characteristic of protein-targeting small organic molecules. Similarly, the knowledge gained in decades of successes and failures in drug discovery campaigns has allowed to compile a list of undesirable chemical features that for several reasons (mostly non-specific interference with biological assays) should not be possessed by drugs, the so-called PAINS (Baell and Holloway, 2010). In structure-based virtual screening, Lipinski's rule of five and PAINS filters are typically applied before the molecular docking. This is to avoid wasting time on performing docking calculations of molecules that will likely be discarded anyway, regardless of their ability to bind to the target.

For RNA targets, the chemical space of small-molecule binders has not been fully characterised yet. Since relatively few small organic molecule binders of RNA targets are known, their expected properties are not yet established and are a hot research topic (Warner *et al.*, 2018; Juru and Hargrove, 2021). Early identified ligands that acted by binding RNA had a positive net charge and were able to intercalate between RNA bases (Thomas and Hergenrother, 2008; Guan and Disney, 2012). However, such physicochemical properties cause non-specific binding on the negatively charged RNA backbone, yielding low selectivity. For this reason, ligands may display a relatively high level of toxicity, therefore often resulting non-viable. Recently, research efforts based on the analysis of ligands with activities towards RNA were directed to the characterisation of the physicochemical space of RNA small-molecule binders (Morgan *et al.*, 2017, 2019; Haniff *et al.*, 2020; Rizvi *et al.*, 2020). The overall picture that is gradually emerging points to an RNA-privileged chemical space. The most

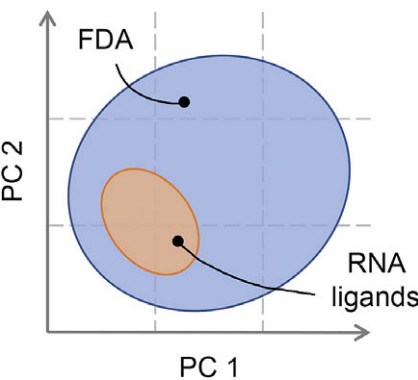

**Fig. 5.** The chemical space of RNA-binding ligands. Bioactive RNA-targeted compounds populate a region in chemical space (here projected along two hypothetical principal components of cheminformatic parameters) occupied by FDA-approved drugs, which mostly target proteins (Juru and Hargrove, 2021). Therefore, while RNA ligands have particular structural and shape properties, they can also possess the typical drug-like properties.

peculiar structural features identified in RNA-targeting ligands appear to be a higher nitrogen count, a lower oxygen count, an enrichment in aromatic rings, fewer stereocenters and fewer sp3-hybridised carbon atoms (Morgan *et al.*, 2019; Haniff *et al.*, 2020). Additionally, in contrast to the more heterogeneous spatial arrangements of the approved drugs, a prevalence of rod- and planar-like shapes (Wirth and Sauer, 2011) has also been observed. Interestingly, we note how, as a whole, these features are reminiscent of the nucleobases. Most remarkably, this RNA-privileged chemical space appears to be a subset of the space traditionally occupied by protein-binding ligands and, more generally, by orally administered drugs (Fig. 5) (Juru and Hargrove, 2021). An interesting implication of this latter aspect is that there is a real potential for these molecules to be RNA-targeting therapeutics. From a more practical perspective, the identified characteristics represent indispensable instruments for computational medicinal chemists to refine screening libraries and to guide the optimisation of promising binders.

While it is essential to take advantage of this knowledge, the number of known RNA binders is still limited, so the exploration and definition of the chemical space of RNA-binding ligands has only just begun. Therefore, further advances may come into play and refine the current picture in the (possibly near) future. Indeed, as a further level of complexity, recently discovered potent and selective ligands of an RNA ribozyme contained chemical groups that would usually be classified as PAINS (Fedorova *et al.*, 2018). This highlights how critical it can be to apply rules of thumb, which were developed to target proteins, for RNA targeting, since the ligands' chemical space is still under construction. Given our limited knowledge, it is therefore important to build screening libraries by taking a non-RNA-biased approach, increasing chemical diversity and maintaining drug-likeness.

In conclusion, we are still in the process of comprehensively characterising the physicochemical properties of RNA binders. Therefore, while the boundaries of their features are gradually becoming clearer, room for potential expansion in this respect should nevertheless be contemplated.

## Outlook and concluding remarks

The growing recognition of RNAs as promising pharmaceutical targets requires a mindset change in drug discovery. In this

Perspective, we discussed the three main challenges of computational RNA-targeted drug discovery: i) the prominent role of target flexibility in predicting small-molecule binding; ii) the importance of achieving binding selectivity; and iii) the knowledge of the chemical space expected for RNA-binding drugs.

We have discussed how the currently available computational procedures, which have been optimised and refined over decades of efforts directed to protein targets, can be used in the novel and partially unexplored context of RNA targets, and how to take appropriate precautions. For example, recent scoring functions were specifically developed to describe the interaction of small molecules with RNA, which means that well-established docking protocols can be easily adapted to RNA targets. Moreover, we discussed how MD simulations, which are extensively used for lead optimisation in protein-based drug discovery, will become an essential tool for considering RNA target flexibility in the earlier stages of screening campaigns. Furthermore, experimental data can increase the reliability of RNA conformational ensembles reconstructed via MD. Interestingly, we note how experimental information often has a dual function when targeting RNA, since the experimental data can be used as the source of starting structures for the simulations and as a guide to refine the conformational ensembles. Moreover, MD simulations (possibly combined with enhanced sampling methods and appropriate analysis tools) may help identify suitable pockets to focus on in the search for specific interactions. In this respect, by showing a greater amount of structural complexity, RNA motifs such as riboswitches, ribozymes, multi-junctions and pseudoknots are intrinsically more inclined to pocket formation, and thus they are better suited for RNA-targeted discovery of small molecule drugs. Given the above and relative to traditional protein-based drug discovery, computational medicinal chemists may need a more skilled background in statistical mechanics and simulative methods in order to take full advantage of these approaches.

Once the chemical space of RNA-targeting drugs has been defined, machine learning and artificial intelligence, which are revolutionising several aspects of conventional drug discovery (Vamathevan et al., 2019), will be critical to developing novel active compounds. Indeed, the most recently developed scoring functions for docking tend to be based on machine learning approaches, and artificial intelligence is already being leveraged to advance the area of force field improvement.

The landscape of currently available small-molecule drugs targeting RNA is somewhat limited. Current drugs are antibiotics targeting ribosomal RNA (rRNA), such as the synthetic oxazolidinone linezolid that acts by binding a highly structured pocket, and the very recent Roche and PTC Therapeutics' risdiplam, used in the treatment of spinal muscular atrophy (SMA), which acts by stabilising the interaction between an RNA splice site and a small nuclear ribonucleoprotein (snRNP) (Sheridan, 2021). Furthermore, Merck's ribocil inhibits bacterial growth by binding to a bacterial riboswitch, however, this small molecule is on hold at the preclinical stage due to the rapid development of bacterial resistance and unlikely will be pursued further (Warner et al., 2018). Despite being limited in number, all these examples support the idea that RNA is a legitimate target of small molecules and highlights the potential of focusing on RNA molecules with high structural complexity to achieve high affinity and selectivity.

Finally, a mention is here required on other types of RNA molecules (not extensively covered in this contribution) as pharmaceutical targets, which are currently in the market/clinical trials for major unmet medical needs. Remarkable examples thereof are the recently approved risdiplam for SMA and the Novartis's branaplam, under clinical trial for both SMA and Huntington disease, which act as splicing modulators by binding to pre-mRNA (Childs-Disney et al., 2022).

In conclusion, time has come for computational drug discovery to embrace the potential of RNA to become an established drug target shortly. Indeed, thanks to the growing interest in discovering small molecules that target RNA, the field is gradually producing useful resources, such as the recent HARIBOSS database of RNA-small molecule structures (Panei et al., 2022). In the same spirit, efforts by the computational community in sharing simulation inputs via dedicated resources (e.g., the PLUMED-NEST initiative (Bonomi et al., 2019)), and sharing computational practises via Jupyter Notebooks (Kluyver et al., 2016), are likely to accelerate the expansion of more complex and sectorial computational skills to successful drug discovery.

**Acknowledgements.** Giovanni Bussi, Angelo Favia and Thorben Fröhlking are gratefully acknowledged for carefully reading the manuscript and providing useful suggestions.

**Open peer review.** To view the open peer review materials for this article, please visit http://doi.org/10.1017/qrd.2022.20.

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
