## [Reviewer Report]

*Comments to Author*: I enjoyed the perspective by Bernetti et al, and consider this topic to be a useful area for discussion by our community. I would recommend that the authors discuss what RNA targeting drugs are currently in use (I think this is only antibiotics which bind in the ribosome), and what new targets might be of interest for the future, and why these are particularly promising, given the very limited number of RNA targeting therapies compared to DNA and proteins. I was also curious as to whether the RNA specific docking tools perform “better” for RNA compared to generic methods, and how this might be quantified. The authors may consider expanding their review to cover these points. There are a couple of minor language and typographical errors, so careful proof-reading is also needed before resubmission. Please could they define their “ADMET” abbreviation in the text for non-medicinal chemists.

---

## [Reviewer Report]

*Comments to Author*: The manuscript by Bernetti et al. provides an overview of the challenges and opportunities in the field of

computational drug discovery of small molecules targeting RNA. This is an emergent field that is attracting a lot of attention

from both the academic and industry perspectives. I found the manuscript concise, yet extremely clear and complete.

The major challenges facing computational researchers are discussed at an appropriate level of detail. I have

just a few comments that I invite the authors to address.

1) The manuscript is focused on non-coding RNA. However, also other types of RNA might be interesting targets.

For example, one of the 3 (to the best of my knowledge) FDA-approved small molecule targeting RNA, Risdiplam,

binds to a pre-mRNA and affects mRNA splicing. I invite the authors to comment on this point.

2) Some of tools that the community developed to target proteins with small molecules have not been extensively (or at all)

tested with RNA targets. For example, I am not sure whether the software mentioned at page 8 to detect binding pockets (SiteMap, NanoShaper, Pocket Finder)

have been used extensively to detect pockets in RNA molecules. Could the authors elaborate on this point?

3) Could the authors give more details bout how the two examples of RNA pockets in Fig. 4 were chosen?

4) Methods to use experimental data to improve the accuracy of MD structural ensembles have been developed by several groups.

I invite the authors to provide at least 2 or 3 additional references (beside Orioli et al, [79], page 7) to provide a more fair

representation of the groups actively working in this field, in particular the Hummer, Vendruscolo and Chodera labs, who are pioneers in the field.

---

## [Reviewer Report]

*Comments to Author*: Reviewer #1: The manuscript by Bernetti et al. provides an overview of the challenges and opportunities in the field of

computational drug discovery of small molecules targeting RNA. This is an emergent field that is attracting a lot of attention

from both the academic and industry perspectives. I found the manuscript concise, yet extremely clear and complete.

The major challenges facing computational researchers are discussed at an appropriate level of detail. I have

just a few comments that I invite the authors to address.

1) The manuscript is focused on non-coding RNA. However, also other types of RNA might be interesting targets.

For example, one of the 3 (to the best of my knowledge) FDA-approved small molecule targeting RNA, Risdiplam,

binds to a pre-mRNA and affects mRNA splicing. I invite the authors to comment on this point.

2) Some of tools that the community developed to target proteins with small molecules have not been extensively (or at all)

tested with RNA targets. For example, I am not sure whether the software mentioned at page 8 to detect binding pockets (SiteMap, NanoShaper, Pocket Finder)

have been used extensively to detect pockets in RNA molecules. Could the authors elaborate on this point?

3) Could the authors give more details bout how the two examples of RNA pockets in Fig. 4 were chosen?

4) Methods to use experimental data to improve the accuracy of MD structural ensembles have been developed by several groups.

I invite the authors to provide at least 2 or 3 additional references (beside Orioli et al, [79], page 7) to provide a more fair

representation of the groups actively working in this field, in particular the Hummer, Vendruscolo and Chodera labs, who are pioneers in the field.

Reviewer #2: I enjoyed the perspective by Bernetti et al, and consider this topic to be a useful area for discussion by our community. I would recommend that the authors discuss what RNA targeting drugs are currently in use (I think this is only antibiotics which bind in the ribosome), and what new targets might be of interest for the future, and why these are particularly promising, given the very limited number of RNA targeting therapies compared to DNA and proteins. I was also curious as to whether the RNA specific docking tools perform “better” for RNA compared to generic methods, and how this might be quantified. The authors may consider expanding their review to cover these points. There are a couple of minor language and typographical errors, so careful proof-reading is also needed before resubmission. Please could they define their “ADMET” abbreviation in the text for non-medicinal chemists.